# Variation-aware Vision Transformer Quantization

## Abstract

Despite the remarkable performance of Vision Transformers (ViTs) in various visual tasks, the expanding computation and model size of ViTs have increased the demand for improved efficiency during training and inference. To address the heavy computation and parameter drawbacks, quantization is frequently studied in the community as a representative model compression technique and has seen extensive use on CNNs. However, due to the unique properties of CNNs and ViTs, the quantization applications on ViTs are still limited and underexplored. In this paper, we identify the difficulty of ViT quantization on its unique **variation** behaviors, which differ from traditional CNN architectures. The variations indicate the magnitude of the parameter fluctuations and can also measure outlier conditions. Moreover, the variation behaviors reflect the various sensitivities to the quantization of each module. The quantization sensitivity analysis and comparison of ViTs with CNNs help us locate the underlying differences in variations. We also find that the variations in ViTs cause training oscillations, bringing instability during quantization-aware training (QAT). Correspondingly, we solve the variation problem with an efficient knowledge-distillation-based variation-aware quantization method. The multi-crop knowledge distillation scheme can accelerate and stabilize the training and alleviate the variation's influence during QAT. We also proposed a module-dependent quantization scheme and a variation-aware regularization term to suppress the oscillation of weights. On ImageNet-1K, we obtain a 77.66% Top-1 accuracy on the extremely low-bit scenario of 2-bit Swin-T, outperforming the previous state-of-the-art quantized model by 3.35%. Code and models will be publicly available.

## 1 Introduction

Vision Transformers (ViTs), inspired by the success of transformer-based models in Natural Language Processing (NLP) tasks, have achieved impressive accuracy on a variety of computer vision tasks (Krizhevsky et al., 2012; He et al., 2016; Tan & Le, 2019). Despite the intrinsic superiority of ViTs, their remarkable performance also comes from the tremendous parameter numbers. For instance, Swin-L (Liu et al., 2021a) of input size $224 \times 224$ has a total number of parameters of 197M with FLOPs of 34.5G. The high latency and large model size have become the most significant obstacle to the efficient deployment of the ViTs, especially on devices with computation constraints.

In recent years, researchers have explored and proposed various model compression methods to improve the computational efficiency of deep learning models. These model compression techniques include quantization (Zhou et al., 2016; Choi et al., 2018; Wang et al., 2019; Esser et al., 2020; Bhalgat et al., 2020; Yamamoto, 2021; Huang et al., 2022), pruning (Liu et al., 2017; 2018; Molchanov et al., 2019; Liu et al., 2019), knowledge distillation (Hinton et al., 2015; Park et al., 2019; Shen & Xing, 2021), and compact network design (Howard et al., 2017; Pham et al., 2018; Guo et al., 2020). Among these methods, quantization of weights and activations have been the most widely utilized techniques because they enjoy the advantage of the promising affinity across different hardware architectures (Judd et al., 2016; Jouppi et al., 2017; Sharma et al., 2018). Although a few efforts (Liu et al., 2021c; Yuan et al., 2021b; Li et al., 2022b; Lin et al., 2021; Li et al., 2022c; Li & Gu, 2022) have been made to apply quantization techniques to ViTs, most of them (Liu et al., 2021c; Yuan et al., 2021b; Lin et al., 2021) are based on Post-Training Quantization (PTQ) which suffers from a significant decline in performance and a bitwidth limitation at 8-bit or 6-bit. Additionally,

the few existing Quantization-Aware Training (QAT) methods (Li et al., 2022b; Li & Gu, 2022; Li et al., 2022a) take much more time than the full-precision model in training, and the models still fail to achieve the desired performance when being quantized to low-precision such as 3-bit and 2-bit.

The lower accuracy of quantized ViTs compared to CNNs guides us to raise the question: *What is it that hinders us from improving the performance of quantized ViTs?* Meanwhile, the low efficiency of previous QAT methods makes applying quantization to more ViT structures difficult. Thus, another question we would like to raise is: *How to improve the efficiency of ViT quantization?*

To comprehensively decipher the inherent obstacles that adversely impact the efficacy and performance of ViT quantization, in this work, we initially conduct an exhaustive investigation of the quantization resilience of each component within the structural layout of the ViTs. The empirical findings derived from the isolated variable (leave-one-out) quantization ablation experiments substantiate that specific constituents, such as Multi-head self-attention (MHSA), exhibit higher sensitivity to quantization compared to other constituents. We further perform a comparative analysis between the weight and activation distribution of ViTs and CNNs, deducing that the intrinsic variability of the distribution serves as the pivotal factor instigating complications with respect to ViTs quantization. This is confirmed through constant monitoring of the weight changing trajectory during the training phase, which revealed that this variability instigates a phenomenon known as weight oscillation. Such a phenomenon has detrimental effects on quantization, potentially culminating in decelerated convergence.

In light of the variation analysis, we propose an optimized solution for ViT quantization that is attuned to variations, demonstrating enhanced efficiency. Initially, a multi-crop knowledge distillation approach is employed, which aids in decreasing the data variance within mini-batches during the training phase, thereby stabilizing and expediting the training process. In terms of the distribution variance observed across differing modules, we introduce a module-specific scaling methodology. This strategy seeks to identify varying scale factors pertinent to different modules, thereby holistically accommodating the diversity in weight distribution through a gradient scaling technique that is sensitive to weight magnitude. When compared with the baseline quantization method, LSQ+ (Bhalgat et al., 2020), the presented approach exhibits less susceptibility to fluctuations in weight distribution and outliers that may arise within ViTs. Furthermore, to combat the potential oscillation throughout the training phase, we

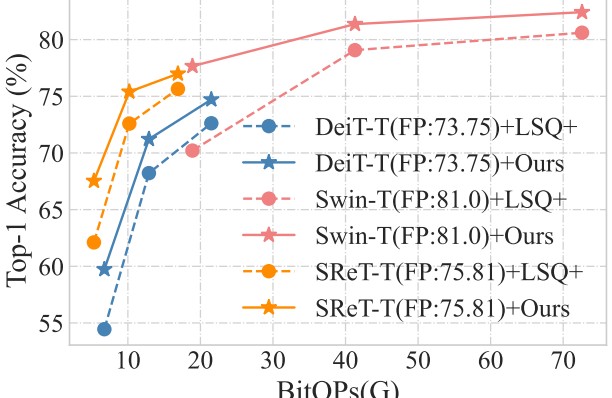

Figure 1: Top-1 accuracy on ImageNet-1K vs. BitOPs comparison of 2/3/4-bit quantized ViT models (DeiT-T, SReT-T, Swin-T) using LSQ+ (Bhalgat et al., 2020) quantization and our method.

put forth a regularization process that is attuned to oscillation within quantization bins. This process seeks to penalize the variance in weight distribution within each respective quantization bin.

Extensive experiments across various ViT architectures with different characteristics, including DeiT (Touvron et al., 2021), Swin Transformer (Liu et al., 2021a), and SReT (Shen et al., 2021a), are conducted to verify the effectiveness and efficiency of our proposed method. For DeiT-T on ImageNet-1K dataset, as shown in Figure 1, our 4-bit quantized model can significantly improve top-1 accuracy to 74.71% compared to the model quantized by LSQ+ (Bhalgat et al., 2020) which achieves 72.62%. Furthermore, to the best of our knowledge, our approach is the first to surpass the full-precision baseline with a 4-bit quantized DeiT-T model and the pioneer in extending the frontier of ViT quantization to a 2-bit level, applicable to both weights and activations. Through these methodologies, we exhibit exceptional training optimization, as evidenced by a 50% reduction in total training duration compared to our established baseline.

In summary, our contribution can be concluded as:

- We reveal the inherent complexity associated with the quantization of ViTs from the perspective of **variation**. Our claims that ViTs grapple with weight fluctuations and activation distribution dispar-

ities are substantiated through sensitivity analysis, comparison of ViTs to CNNs, and investigation of oscillatory behavior.

- We adopt a multi-crop knowledge distillation-based quantization methodology to decrease the data variance within mini-batches during training following (Shen & Xing, 2021), and introduce module-dependent quantization and oscillation-aware regularization strategies. The proposed method is capable of mitigating the impact of variations in ViTs.

- We perform extensive experiments on DeiT, Swin, and SReT architectures using the ImageNet-1K dataset. Our approach significantly outperforms prior state-of-the-art quantization schemes, demonstrating both superior efficiency and performance.

## 2 Related Work

**Vision Transformer:** Transformer (Vaswani et al., 2017) was originally proposed for natural language processing tasks and demonstrated remarkable performance across various benchmarks. Inspired by the success, Vision Transformers(ViTs) (Dosovitskiy et al., 2020) utilize multi-head self-attention blocks for replacing convolutions and treating an image as patches/tokens. The attention mechanism can help capture both short-range and long-range visual dependencies. DeiT (Touvron et al., 2021) introduced a teacher-student distillation token strategy and employed various data augmentation techniques in the training of ViTs and significantly improved the effectiveness and efficiency. Swin (Liu et al., 2021a) proposed the shift window attention scheme at various scales to limit the self-attention computation in local windows, which largely boosts the performance and reduces complexity. Recently, SReT (Shen et al., 2021a) has been proposed with a weight-sharing mechanism by a sliced recursion structure. The convolution layers in SReT also help supplement the inductive bias lacking in ViTs. Various extensions of ViTs (Wu et al., 2021; Yuan et al., 2021; Dong et al., 2022) and more applications (Zheng et al., 2021; Caron et al., 2021; Bertasius et al., 2021; Arnab et al., 2021; Wang et al., 2021) are still emerging.

**Quantization Techniques:** Quantization techniques aim to replace the full-precision weights and activations with lower-precision representation. Based on the quantization intervals, they can be categorized into uniform and non-uniform quantization. While uniform quantization (Zhou et al., 2016; Choi et al., 2018; Esser et al., 2020) with uniform quantization interval has better hardware affinity and efficiency, Non-uniform quantization (Miyashita et al., 2016; Zhang et al., 2018; Li et al., 2019), due to its flexible representation, can usually better allocate the quantization values to minimize the quantization error and achieve better performance than uniform schemes. In addition, the quantization methods can also be classified as quantization-aware training (QAT) (Zhou et al., 2016; Esser et al., 2020; Bhalgat et al., 2020) and post-training quantization (PTQ) (Nagel et al., 2020; Fang et al., 2020; Wang et al., 2020) based on whether to retrain a model with quantized weights and activations or start with a pre-trained model and directly quantize it without extra training. The majority of previous ViT quantization methods, such as Liu *et al.* (Liu et al., 2021c), PTQ4ViT (Yuan et al., 2021b), and FQ-ViT (Lin et al., 2021), focused on PTQ of ViTs. Due to the intrinsic restriction of PTQ, these methods only perform 8-bit or 6-bit quantization.

**Knowledge Distillation:** The concept of knowledge distillation is first proposed in (Hinton et al., 2015), where the core insight is to encourage student models to emulate the distribution of teacher models' prediction. The prediction distribution of teacher models contains more information than the one-hot labels. More recently, various knowledge distillation methods (Cho & Hariharan, 2019; Park et al., 2019; Tung & Mori, 2019; Mirzadeh et al., 2020; Shen & Xing, 2021) have been proposed for better efficiency and effectiveness. The knowledge-distillation methods are also widely adopted in previous research (Mishra & Marr, 2018; Polino et al., 2018; Huang et al., 2022) to help quantization-aware training.

## 3 Approach

### 3.1 ViT Architecture and Quantization

**ViT Architecture.** The basic block of ViTs is the transformer layer, consisting of Multi-head Self Attention (MHSA), Layer Normalization (LN) (Ba et al., 2016), and Feed-forward Network (FFN). The transformer

layer can be formulated as:

$$\mathbf{X}' = \text{LN}(\mathbf{X_i} + \text{MHSA}(\mathbf{X_i}))$$
$$\mathbf{X_O} = \text{LN}(\mathbf{X}' + \text{FFN}(\mathbf{X}')), \tag{1}$$

where $\mathbf{X_i}$, $\mathbf{X}'$, and $\mathbf{X_o}$ are this transformer block's input, intermediate representation, and output. The MHSA module consists of $h$ heads, and each head performs inner products with a scaling factor and a *softmax* operation. For the $i$-th head, input $\mathbf{X_i}$ is projected into *query*, *key*, and *value* vectors with multiplication with learnable weight matrix $\mathbf{W_{Q,i}}, \mathbf{W_{K,i}}, \mathbf{W_{V,i}}$ respectively, which can be written as:

$$\mathbf{Q_i} = \mathbf{X_i W_{Q,i}}, \mathbf{K_i} = \mathbf{X_i W_{K,i}}, \mathbf{V_i} = \mathbf{X_i W_{V,i}}, \tag{2}$$

and the output of $i$-th head is

$$\text{head}_\mathbf{i} = \text{softmax}(\mathbf{Q_i K_i^T}/\sqrt{\mathbf{d_k}})\mathbf{V_i}, \tag{3}$$

where $1/\sqrt{\mathbf{d_k}}$ is the scaling factor for normalization. MHSA further concatenates the output of these heads to improve the representative capacity and projects to the output by multiplication with a learnable weight matrix $\mathbf{W_o}$:

$$\text{MHSA}(\mathbf{X_i}) = \text{Concat}(\text{head}_\mathbf{1}, \text{head}_\mathbf{2}, ..., \text{head}_\mathbf{h})\mathbf{W_o}. \tag{4}$$

**Quantization.** Given the real-value data to be quantized as $x^r$, the scale factor $s$ of the quantizer, the number of positive quantization levels $Q_P$, and the number of negative quantization levels $Q_N$, we can have the quantizer $q_b$ that output the $b$-bit quantized representation of the input real value as $x^q = q_b(x^r)$ :

$$x^q = q_b(x^r) = s \times \lfloor \text{clip}(x^r/s, -Q_N, Q_P) \rceil, \tag{5}$$

where $\lfloor \cdot \rceil$ is the rounding function that rounds the input to the nearest integer, $\text{clip}(x, r_1, r_2)$ return $x$ with all value below $r_1$ set to be $r_1$ and all values above $r_2$ set to be $r_2$. For the unsigned quantization, $Q_N = 0, Q_P = 2^b - 1$. While for the quantization of signed data, $Q_N = 2^{b-1}, Q_P = 2^{b-1} - 1$. To solve the problem that the gradient cannot back-propagate in Equation 5, the straight-through estimator (STE) (Bengio et al., 2013) is utilized to approximate the gradient during quantization-aware training. The gradient of the rounding operation is approximated as 1 in the quantization limit. In the back-propagation with STE, the gradient of the loss $\mathcal{L}$ with respect to the real-value data $x^r$ is set to be:

$$\frac{\partial \mathcal{L}}{\partial x^r} = \frac{\partial \mathcal{L}}{\partial x^q} \cdot \mathbf{1}_{-Q_N \le x^r/s \le Q_P}, \tag{6}$$

where $\mathbf{1}$ is the indicator function that outputs 1 within the quantization limit and 0 otherwise. This STE is widely used in quantization-aware training (QAT). Correspondingly, we focus on uniform quantization and QAT in this work.

### 3.2 Understanding Variation of ViTs

Many existing studies highlight that ViTs exhibit greater sensitivity to quantization compared to CNNs. For instance, Bit-Split (Wang et al., 2020), which successfully achieves 4-bit quantization on ResNet with an accuracy loss of less than 1%, exhibits significant accuracy degradation of over 2% (Lin et al., 2021) when applied to 8-bit quantization of DeiT. However, there is a paucity of comprehensive analyses detailing the reasons behind ViTs' heightened computational sensitivity compared to CNNs. In this section, we will primarily examine the quantization sensitivity of each component via a leave-one-out quantization analysis. Upon identifying the problematic areas or "pain points" in ViT quantization, we will contrast ViTs with CNNs to validate the fundamental challenge in quantization, referred to in this work as **variation**. We define the term **variation** to include two components: (1) the differential sensitivity and importance of each module and (2) the variance of weight distribution. We will explore the variation in sensitivity in Section 3.2.1 and delve into the variation in distribution and its subsequent side-effect of oscillation phenomenon in Sections 3.2.2 and 3.2.3.

### 3.2.1 Quantization Sensitivity Analysis

Prior study Q-ViT (Li et al., 2022b) conducted a quantization robustness analysis on ViTs, concluding that the GELU activation function substantially mitigates performance during the quantization process. However, their experiments relied on post-training quantization (PTQ), which stands in stark contrast to quantization-aware training (QAT). Moreover, their experimental methodology lacked a comprehensive analysis of different components at a more granular level, such as the quantization impact on query, key, and value weight matrices. In this section, we aim to disentangle the intricacies of ViT quantization by executing an in-depth leave-one-out analysis employing QAT.

In terms of quantization methods, we employ LSQ+(Bhalgat et al., 2020). All components except for the analysis target will be quantized to 3-bit, while the analysis target will be retained at full precision. The experimental results using DeiT-T on the ImageNet-1K are presented in Table 1. These results indicate that MHSA, particularly the *value* weight matrices, are highly susceptible to quantization. Although MHSA and the *value* weight matrix constitute a relatively minor fraction of parameters in comparison to the FFN, maintaining these parts at full precision can optimize the performance of the quantized model.

Table 1: Leave-one-out-anlysis for quantization of various components in DeiT-T on ImageNet-1K. The Para(%) stands for the percentage of parameters that are **not** quantized among all trainable parameters.

| Quantization Target | Top-1 Acc(%) | Top-5 Acc(%) | Para(%) |
|---|---|---|---|
| None (FP Model) | 73.75 | 91.87 | 100 |
| All (Baseline 3-bit) | 68.22 | 88.56 | 0 |
| All, except FFN | 69.47 | 89.60 | 62.1 |
| All, except MHSA | **71.28** | **90.66** | 31.1 |
| All, except *query* in MHSA | 69.66 | 89.94 | 7.8 |
| All, except *key* in MHSA | 69.92 | 89.81 | 7.8 |
| All, except *value* in MHSA | **70.72** | **90.40** | 7.8 |

While we have fully exploited the clue that the quantization sensitivity of MHSA is higher than other components in ViTs, another critical clue is that some heads in MHSA are more important than other heads in Transformer-based models, which has already been proved in NLP tasks (Michel et al., 2019). Here we apply a similar analysis as (Michel et al., 2019) to quantize various heads in different layers in ViTs. The target heads are quantized to 2-bit while the remaining components are quantized to 8-bit. The results of DeiT-T with three heads in a layer and 12 layers are shown in Figure 2.

Figure 2: The accuracy degradation compared to the full-precision model when a specific head in a layer is quantized. The label *h-l* in abscissa indicates the head *h* in layer *l* is quantized.

The results that some heads have higher accuracy degradation show that the quantization sensitivity of different heads at different layers varies. The first and last few layers are more sensitive to quantization. Additionally, the heads in the same layer also show a quantization robustness variation. For example, in layer 8 of the quantized model, the lower precision of head 0 (shown in 8-0 in Figure 2) will result in higher accuracy drop compared to the two parallel heads in the same layer.

### 3.2.2 Variation of ViTs and CNNs

In Section 3.2.1, we have demonstrated that ViTs suffer from significant variation in the sensitivity to quantization. However, previous mixed precision quantization research on CNN has also discovered that different parts of models have various quantization robustness. To fully understand why the sensitivity to quantization in ViTs is higher than CNNs, we visualize and quantify the distribution of different modules inside full-precision CNNs and ViTs to compare the real **variation** of ViT and CNN models.

To give an intuitive result on the variation of CNNs and ViTs, we first visualize the weight distribution across different channels in pre-trained full precision ResNet-18 (He et al., 2016) and DeiT-T. The results

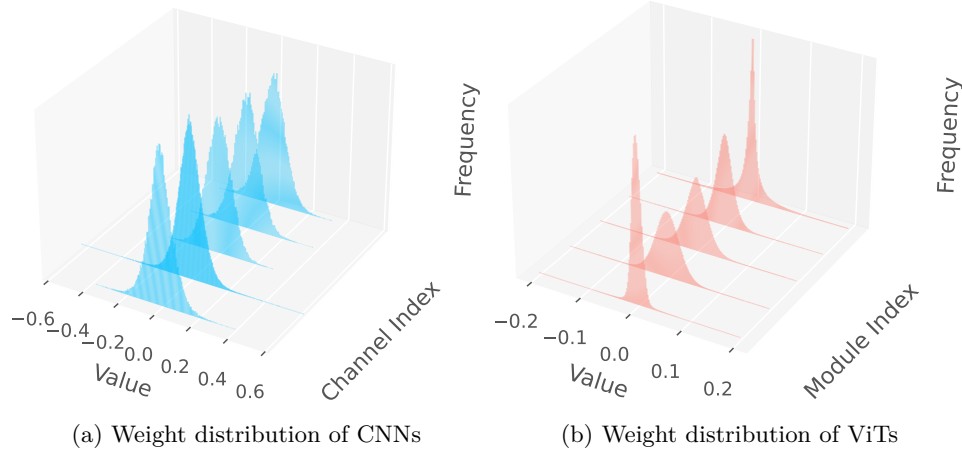

(a) Weight distribution of CNNs           (b) Weight distribution of ViTs

Figure 3: The weight distribution variance in CNNs (ResNet-18) and ViTs (DeiT-T). The visualized weight tensors are randomly selected from different channels in CNNs and different modules (heads) in ViTs.

are shown in Figure 3. Based on our investigation, the ResNet-18 model shares a similar distribution across different channels, while the weight distribution varies significantly in different modules in DeiT-T.

To quantify the fluctuation in the latent real-valued weight magnitude, we proceed to calculate the Average Standard Deviation of the Absolute Mean (SDAM) of the real-valued weight magnitude within each module of CNNs and ViTs. The SDAM metric has been previously employed to evaluate the stability and fairness

Table 2: Standard Deviation of the Absolute Mean (SDAM) of real-value weight in CNNs and ViTs.

| Model | ResNet-18 | VGG-11 | ViT-T | DeiT-T | Swin-T |
|---|---|---|---|---|---|
| SDAM | 5.59e-2 | 3.74e-2 | 9.65e-2 | 8.35e-2 | 9.71e-2 |

of training in prior studies (Liu et al., 2021b). The corresponding results of the SDAM comparison are tabulated in Table 2. These numerical findings corroborate that the variability associated with ViTs surpasses that of CNNs with respect to the weight distribution.

Correspondingly, prior work (Lin et al., 2021) has highlighted significant disparities in the distribution of activations in ViTs as opposed to CNNs. Although these variations may augment the representational capacity of ViTs, they concurrently introduce complexities when implementing quantization in the context of ViT models. Consequently, the conception and arrangement of the quantization scheme become paramount, particularly in the generation of quantization scales and the determination of clipping factors during the process of quantization-aware training.

### 3.2.3 Oscillation in Training

High variance in weight and activation distribution can lead to suboptimal quantization, thereby inducing increased quantization errors. In quantization-aware training, certain modules fail to learn meaningful representation during the optimization process. This effect and its association with distribution variation have been investigated in AdamBNN (Liu et al., 2021b), where the notion of *flip-flop* was introduced, signifying the change in quantization results of weights at specific iterations. We observed that low-precision quantization of ViTs is also subject to a comparable effect, termed **oscillation**. This denotes the circumstance where the latent weights fluctuate around the boundary of adjacent quantization bins during quantization-aware training. As per our understanding, (Nagel et al., 2022) is the sole work probing into these effects, however, it restricts its scope to CNNs and their impact on batch normalization, a technique not employed in ViTs. We take the initiative to identify and analyze this oscillation phenomenon specific to ViTs.

An illustration of the oscillation phenomenon is shown in Figure 4. Conventionally, the distribution of full-precision initialization adheres to a Gaussian distribution. There exist only a limited number of latent weights that precisely coincide with the optimal quantization value. A majority of weights necessitate updates

during the process of quantization-aware training. However, when certain real-value weights $w_t^r$ cross the quantization boundary at a particular iteration $t$, the update of real weights $|w_t^r - w_{t-1}^r|$ triggers an update in the quantized value by a constant value $|q(w_t^r) - q(w_{t-1}^r)| = s$. Here, $s$ represents the quantization scale and constitutes the length of a quantization bin within the framework of a uniform quantization scheme. As indicated by the STE detailed in Equation 6, the gradient of the real value is assigned a value identical to this quantized value, resulting in a consistent gradient that encourages the real value to once again traverse the quantization boundary, given that the learning rate remains consistent.

We further observe the side effect in the quantization-aware training of ViTs. As shown in Figure 4a, the weights associated with MHSA tend to accumulate around the quantization threshold following a certain number of training epochs. Figure 4b presents an example of this oscillatory behavior within the weights of ViTs. This oscillation effect adversely influences the training of ViTs and leads to substantial quantization error. The formulation of a solution to prevent this phenomenon, through the reduction of variation and mitigation of the impact, will be central to our design methodology for quantization.

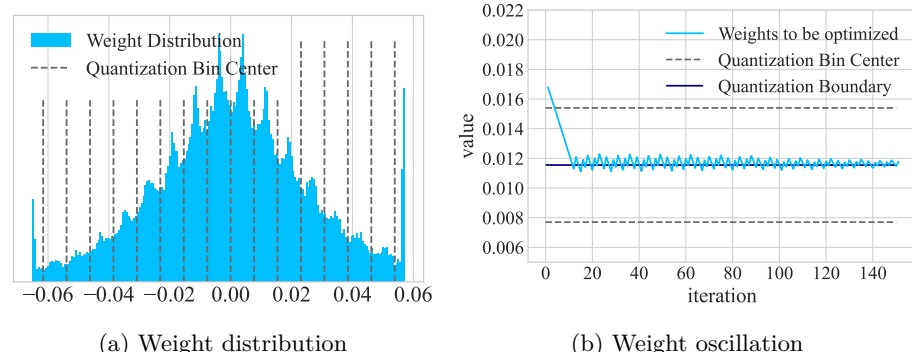

(a) Weight distribution          (b) Weight oscillation

Figure 4: The visualization of weight distribution during quantization-aware training and the weight oscillation effect due to distribution variance. The layer we select is *blocks.1.attn.proj-v.weight* in 4-bit quantized DeiT-S with scale $\alpha = 0.0077$.

### 3.3 Variation-aware ViT Quantization

As observed in Section 3.2, there exists a substantial fluctuation amongst all components of ViTs, which can precipitate an oscillation phenomenon potentially introducing instability during training. Motivated by this observation, we aim to introduce a **variation-aware** quantization scheme to mitigate the impacts of such fluctuations and enhance both the effectiveness and computational efficiency of ViT quantization. As illustrated in Figure 5, our approach incorporates several crucial components: training facilitated by multi-crop knowledge distillation, a module-specific quantization scheme, and a regularization strategy sensitive to oscillatory behavior.

#### 3.3.1 Multi-crop Knowledge Distillation

To solve the variation mentioned in Section 3.2.2 and help stabilize the training, we first propose a Multi-crop Knowledge Distillation (MCKD) scheme. The core insight is to train our quantized ViT models with a full-precision model as the teacher. The loss function is designed to enforce the similarity between the output distribution of the full-precision teacher and quantized student ViT model:

$$\mathcal{L}_{\text{Vanilla}KD} = -\frac{1}{N} \sum_c \sum_{i=1}^N p_c^{T_f}(X_i) \log(p_c^{S_q}(X_i)), \tag{7}$$

where the KD loss is defined as the cross-entropy between the output distributions $p_c$ of a full-precision teacher $T_f$ and a quantized ViT student $S_q$. $X_i$ is the input sample. $c$ and $N$ denote the classes and the number of samples, respectively. Note that one-hot label is not involved in training in our setting. The KD scheme helps our model converge fast because it learns the mapping directly from the full-precision teacher, which contains richer information. Previous research (Yuan et al., 2020; Zhou et al., 2020; Menon et al.,

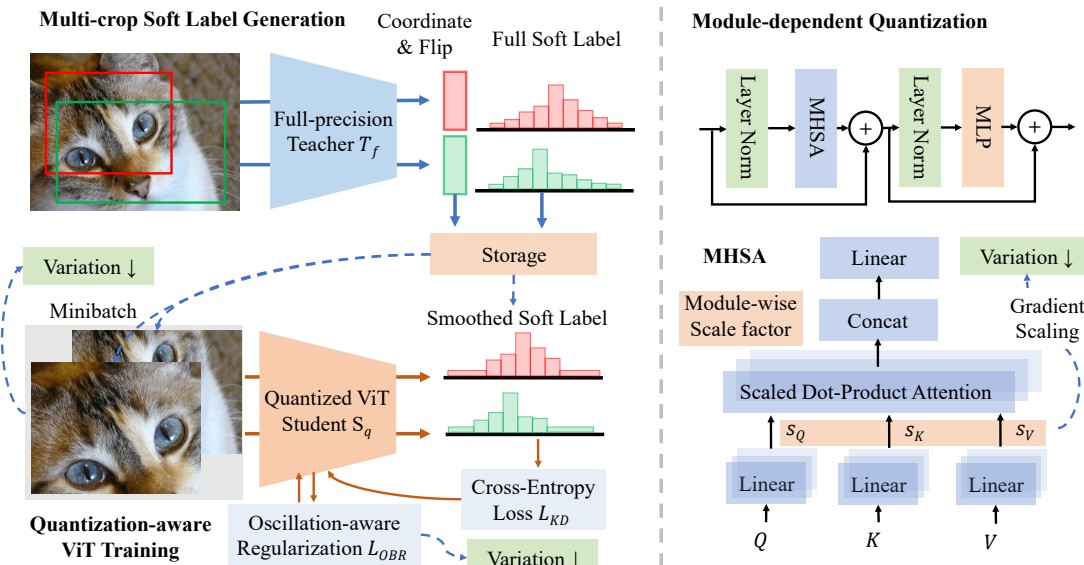

Figure 5: An overview of our efficient variation-aware quantization method. The left part illustrates how we perform QAT with multi-crop knowledge distillation. The right part demonstrates the proposed module-dependent quantization scheme.

2021) also points out that KD loss can be seen as a regularization term to reduce the variance during the training, which makes the training more stable and alleviates the influence of the distribution variation. However, here we only employ KD loss as the sole objective to optimize the target model, which has been demonstrated more effective with adequate supervision signal in KD (Shen et al., 2021b).

One disadvantage of the conventional KD training scheme is that generating the prediction $p_c^{T_f}$ of the teacher $T_f$ consumes a relatively long time, which makes the training inefficient. To tackle this challenge, we propose to use a multi-crop KD scheme as FKD that first random crops $M$ regions from one image $X_i$, and inputs each cropped image to the teacher model $T_f$ to get the soft label $p_c^{T_f}(X_{i,m}), m \in M$, where $m$ is the index of the cropped region. The soft label is stored together with its coordinates and augmentation hyper-parameters. In the training phase, we directly load the soft label and cropping parameter from the storage, and the cropped sample used for the training with KD. The loss function of this multi-crop KD scheme is:

$$\mathcal{L}_{KD} = -\frac{1}{NM} \sum_c \sum_{i=1}^{N} \sum_{m=1}^{M} p_c^{T_f}(X_{i,m}) \log(p_c^{S_q}(X_{i,m})). \tag{8}$$

The higher quality of the soft label generated by this scheme would reduce the variation within a mini-batch to a greater extent. Meanwhile, the data and its corresponding label is loaded the same as the training without knowledge distillation, where the time for inference with the teacher model is saved. We further show in the experiment that this multi-crop KD scheme improves performance by reducing variation and significantly boosts efficiency.

### 3.3.2 Module-dependent Quantization

We utilize the same scale learning strategy as LSQ+ (Bhalgat et al., 2020), wherein the scale factor $s$ is dynamically learned during the optimization. Our exploration in Section 3.2.1 establishes a substantial variation in the sensitivity of distinct modules to quantization. However, conventional implementations of ViT quantization often overlook this characteristic. In view of the variability observed in ViTs, we propose a module-dependent quantization scheme that facilitates the learning of the quantization scale $s$ at the granular module level (*query*, *key*, and *value* in distinct heads of MHSA). This approach contrasts with previous layer-wise quantization methods that assigned a uniform scale to differing modules. Instead, we implement scale-learning quantization at a higher resolution, thereby promoting a finer granularity.

Previous work (Bhalgat et al., 2020) has pointed out the negative impact of an imbalance gradient scale. However, the situation is even more severe in the quantization of ViTs, as weight distribution shows a significant variation. To overcome this challenge, we adopt a module-dependent gradient scaling that balances the weights and scale factor gradient, fully considering the distribution variation in different modules. We multiply the loss of scale factor $s$ by a gradient scale $g$ that encodes the magnitude of the weights in this module, which can be formulated as $\frac{\partial \mathcal{L}}{\partial s} \longleftarrow \frac{\partial \mathcal{L}}{\partial s} \cdot \frac{1}{\sqrt{Q_P ||w||_1}}$, where $||w||_1$ computes the $L_1$-norm of weights in the quantized module. For the modules with higher variation, the $L_1$-norm of weights will be higher than average, and the update of scale factor $s$ will be decreased to ensure that the outliers of the distribution do not influence the scale factor.

### 3.3.3 Oscillation-aware Bin Regularization

In the analysis of Section 3.2.3, we identify that the weight distribution variance in ViTs caused oscillation, leading to instability during training. In the view of distribution in each quantization bin, the majority of the weights oscillate between both sides of the quantization bin. To suppress the oscillation phenomenon during QAT, we regularize the weight distribution with an Oscillation-aware Bin Regularizer (OBR) to encourage the real-value weights to be close to the quantization bin center. The proposed OBR can be formulated as

$$\mathcal{L}_{OBR} = \sum_{m=1}^{M} (||w_m^r - w_m^q||_2 + \sum_{n=1}^{2^b} \mathcal{V}(w_{n,m}^r)), \tag{9}$$

where $w_m^r, w_m^q, w_{n,m}^r$ represent the real value and quantized value of weights in module $m$, and real value weights in the quantization bin $n$, respectively. $|| \cdot ||_2$ computes the $L_2$-norm and $\mathcal{V}(\cdot)$ computes variance for all quantization bins with more than two elements.

Unlike the previous weight regularization (Chmiel et al., 2020) applied in quantization which only considers the global weight distribution, we minimize the global quantization error and local distribution variance in a specific quantization bin. Ideally, the distribution of the weights in a quantization bin is regularized to be a Dirac delta distribution which can largely suppress the oscillation during training. The final optimization target is $\mathcal{L} = \mathcal{L}_{KD} + \lambda \mathcal{L}_{OBR}$, where $\lambda$ is the weighting coefficient to balance between $\mathcal{L}_{KD}$ and $\mathcal{L}_{OBR}$. To make sure that the regularization does not influence the learning of scale factors at the very early stage of training, we gradually increase the coefficient $\lambda$ during training by applying a cosine annealing schedule following (Nagel et al., 2022).

Table 3: Comparison with previous quantization methods on ImageNet-1K. "Bit-width (W/A)" denotes the bitwidth for weights and activations. "Epochs" denote the total training epochs.

| Network | Method | Epochs | FP Top-1 | Bit-width (W/A) | Top-1 | Bit-width (W/A) | Top-1 | Bit-width (W/A) | Top-1 |
|---|---|---|---|---|---|---|---|---|---|
| DeiT-T | Q-ViT (Li et al., 2022b) | 300 | 72.86 | 4/4[†] | 72.79 | 3/3[†] | 69.62 | - | - |
| | LSQ+ (Bhalgat et al., 2020) | 300 | 73.75 | 4/4 | 72.62 | 3/3 | 68.22 | 2/2 | 54.45 |
| | LSQ+ w/ KD | 300 | 73.75 | 4/4 | 73.56 | 3/3 | 69.83 | 2/2 | 56.29 |
| | Ours | **150** | 73.75 | 4/4 | **74.71** | 3/3 | **71.22** | 2/2 | **59.73** |
| SReT-T | LSQ+ (Bhalgat et al., 2020) | 300 | 75.81 | 4/4 | 75.65 | 3/3 | 72.59 | 2/2 | 62.11 |
| | LSQ+ w/ KD | 300 | 75.81 | 4/4 | 76.13 | 3/3 | 74.20 | 2/2 | 64.98 |
| | Ours | **150** | 75.81 | 4/4 | **76.99** | 3/3 | **75.40** | 2/2 | **67.53** |
| Swin-T | Q-ViT (Li et al., 2022b) | 300 | 80.9 | 4/4[†] | 80.59 | 3/3[†] | 79.45 | - | - |
| | LSQ+ (Bhalgat et al., 2020) | 300 | 81.0 | 4/4 | 80.61 | 3/3 | 79.07 | 2/2 | 70.21 |
| | LSQ+ w/ KD | 300 | 81.0 | 4/4 | 81.37 | 3/3 | 80.01 | 2/2 | 73.50 |
| | Li et al. (Li et al., 2022a)* | 300 | 81.0 | 4/4 | 82.10 | 3/3 | 80.57 | 2/2 | 74.31 |
| | Ours | **150** | 81.0 | 4/4 | **82.42** | 3/3 | **81.37** | 2/2 | **77.66** |

[†] average bitwidth for mixed-precision quantization
[*] our implementation with the same full-precision model as initialization

# 4 Experiments

## 4.1 Experimental Settings

**Dataset** The experiments are carried out on the ImageNet-1K dataset (Deng et al., 2009). We only perform basic data augmentation in PyTorch (Paszke et al., 2019), which includes *RandomResizedCrop* and *RandomHorizontalFlip* during the training and single-crop operation during the evaluation.

**Model** We evaluate our quantization methods on three ViT architectures: DeiT-T (Touvron et al., 2021), SReT-T (Shen et al., 2021a), and Swin-T (Liu et al., 2021a). Due to the fact that the first (patch embedding) and the last (classification) layer are more sensitive to quantization perturbation compared to intermediate layers, we fix their bitwidth to 8-bit following previous work (Yang & Jin, 2021).

**Training Detail** Following previous quantization methods (Zhou et al., 2016), we adopt real-value pre-trained weights as initialization. The quantization parameters, including scale factors and offset, are initialized using the MSE-based method following (Bhalgat et al., 2020). Details of all hyper-parameters and training schemes are shown in the Appendix.

## 4.2 Comparison with State-of-the-Art Methods

Table 3 compares our efficient variation-aware quantization with existing methods for DeiT-T, SReT-T, and Swin-T on the ImageNet-1K dataset. As we utilize different full-precision (FP) models as initialization, the corresponding FP Top-1 accuracy is also reported. To show that our performance improvement cannot simply be summarized as learning from the large teacher model, we also report the results of LSQ+ with vanilla knowledge distillation using the same teacher model. Compared with the baseline FP mode, our 4-bit quantized DeiT-T achieves 74.71% Top-1 accuracy, which is the **first 4-bit quantized model** with accuracy higher than FP initialization (0.96% absolute gain). Similarly, our 4-bit quantized SReT-T and Swin-T achieve 76.99% and 82.42% Top-1 accuracy, which is 1.18% and 1.42% higher than the FP baseline.

Compared with the previous quantization methods LSQ+ (Bhalgat et al., 2020), mixed-precision method Q-ViT (Li et al., 2022b), and state-of-the-art (Li et al., 2022a), our model also demonstrates remarkable improvement. For example, our 4-bit Swin-T achieves a Top-1 accuracy of 82.42%, which has an absolute gain of 2.83% compared to Q-ViT (Li et al., 2022b). Our method is especially effective for low-precision 2-bit quantization, as our 2-bit quantized Swin-T yields 77.66% Top-1 accuracy, which is 3.35% higher than the previous state-of-the-art method (Li et al., 2022a).

In addition, our methods show better efficiency with the help of a multi-crop knowledge distillation scheme. The better quantization scheme and regularization also help our models converge faster than previous methods with the same training configurations. We only train our models with 150 epochs, sufficient to outperform previous methods in terms of accuracy. The total training time for our DeiT-T with 4 NVIDIA A100 GPUs is 57.3 hours, significantly lower than baseline methods shown in Table 5.

## 4.3 Ablation Study

We first perform an overall ablation experiment on 4-bit quantized DeiT-T to look into the effectiveness of all proposed modules. The results are shown in Table. 4. From the average Standard Deviation of the Absolute Mean (SDAM) and accuracy results, we can see that each module helps alleviate the variation influence and improve the perfor-

Table 4: Overall ablation experiment on 4-bit quantized DeiT-T. For the experiment "Ours w/o MCKD", the vanilla knowledge distillation with a ResNet152 teacher is applied.

| Method | Top-1 Acc | Top-5 Acc | SDAM |
|---|---|---|---|
| Ours | 74.71 | 92.02 | 2.13e-2 |
| Ours w/o Multi-crop Knowledge Distillation | 73.56 | 91.52 | 2.30e-2 |
| Ours w/o Module-dependent Quantization | 73.79 | 91.54 | 7.15e-2 |
| Ours w/o Oscillation-aware Bin Regularization | 74.22 | 91.41 | 3.79e-2 |

mance of quantized ViTs. The following subsections give a more detailed ablation study on each module.

**Multi-crop Knowledge Distillation** Table 5 compares the Top-1 accuracy of 4-bit quantized DeiT-T without knowledge distillation, with vanilla KD, and with our multi-crop KD of different teachers. The results demonstrate an improvement in both accuracy and efficiency. The teacher model of higher accuracy can improve the performance of student ViTs regardless of architecture. The training time can also be reduced as the soft label is extracted before the training. The time in Table 5 does not include the time for soft label generation, which can be ignored when we have to apply QAT on different models and settings.

Table 5: Comparison of different teacher models of knowledge distillation for our 4-bit quantized DeiT-T on ImageNet-1K. "Training Time" indicates the GPU hours of the training process on 4 NVIDIA A100 GPUs.

| Method | Teacher | Top-1 Acc | Top-5 Acc | Training Time (h) |
|---|---|---|---|---|
| Ours w/o KD | Ground Truth | 72.62 | 91.19 | - |
| Ours w/ Vanilla KD | ResNet152 (He et al., 2016) | 73.56 | 91.52 | 143.5 |
| Ours w/MCKD | ResNet152 (He et al., 2016) | 74.26 | 91.81 | **57.3** |
| | BEiT-L (Bao et al., 2021) | 74.49 | 91.92 | |
| | EfficientNet-L2 (Xie et al., 2020) | **74.71** | **92.02** | |

**Module-dependent Quantization** The proposed module-dependent quantization applies a finer-grained quantization scheme at the module level and scales the scale factors' gradients to ensure the scale factor update is not influenced by the variation in ViTs. We visualize the loss landscape showing the smoothness of optimization following (Li et al., 2018) shown in Figure 8b. Compared to the baseline quantized model, the more centralized and smoother loss landscape reflects that the proposed quantization scheme substantially improves the training stability and efficiency.

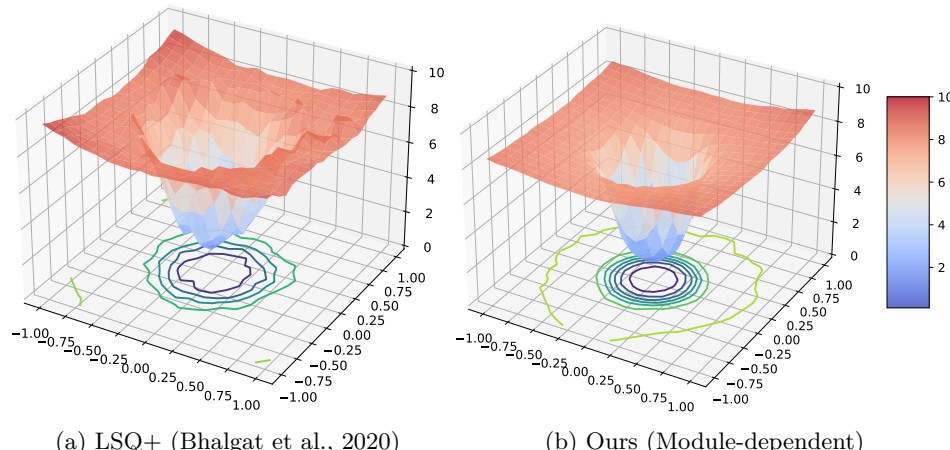

(a) LSQ+ (Bhalgat et al., 2020)    (b) Ours (Module-dependent)

Figure 6: Loss landscape visualization of the 4-bit quantized Swin-T using the baseline (LSQ+ quantization) method and our module-dependent quantization method.

**Oscillation-aware Bin Regularization** To better know how our oscillation-aware bin regularization can help alleviate the oscillation, we quantify the degree of oscillation during training by measuring the frequency of this phenomenon over time. We define that the oscillation occurs at iteration $t$ when the quantized integer value changes and the direction of the update in integer value also changes. This can be formulated as:

$$x_t^{\text{int}} \neq x_{t-1}^{\text{int}}, \text{sign}(\Delta_{\text{int}}^t) \neq \text{sign}(\Delta_{\text{int}}^{t^{\text{prev}}}), \tag{10}$$

where $x_t^{\text{int}} = \lfloor \text{clip}(x^r/s, -Q_N, Q_P) \rceil$ is the integer value of input real-value $x^r$ following the notion in Equation 5. The update $\Delta_{\text{int}}^t = x_t^{\text{int}} - x_{t-1}^{\text{int}}$ and $t^{\text{prev}}$ is the iteration of last integer value change. Then the frequency of oscillation is measured using an exponential moving average (EMA):

$$f^t = m \cdot \text{sign}(\Delta_{\text{int}}^t) + (1-m) \cdot f^{t-1}. \tag{11}$$

We define the weights as oscillating weights at iteration $t$ as $f^t >0.005$. The Top-1 Accuracy of 3-bit quantized SReT-T and the percentage of oscillating weights are shown in Table 6. From the results, we can see a clear negative correlation between weight oscillation percentage and model performance. The proposed Oscillation-aware Bin Regularization (OBR) with a gradually increasing coefficient helps stabilize the training to achieve higher model accuracy.

### 4.4 Attention Map Visualization

To demonstrate how our quantization approach preserves the representational capacity of ViT models, we illustrate the attention map of the quantized Swin-T following (Dosovitskiy et al., 2020) and (Abnar & Zuidema, 2020). We fuse the attention heads utilizing maximum operators and exclude low attention pixels to better accentuate the prominent object within the image. As shown in Figure 7, our quantized Swin-T exhibits superior representational capacity by maintain-

Table 6: Comparison of 3-bit quantized SReT-T using different regularization. "Oscillation" indicates the percentage of weights that are oscillated at the last iteration of training.

| Regularization | Top-1 Acc | Top-5 Acc | Oscillation (%) |
|---|---|---|---|
| Baseline | 75.02 | 92.31 | 7.33 |
| KURE (Chmiel et al., 2020) | 74.85 | 92.24 | 8.12 |
| Ours $\lambda=\cos(0,1)$ | 75.06 | 92.32 | 0.23 |
| $\lambda=\cos(0,0.1)$ | **75.40** | **92.49** | 0.78 |
| $\lambda=\cos(0,0.01)$ | 75.11 | 92.36 | 4.36 |

ing a more relative ranking within the attention map. This distinction becomes more pronounced when the ViT model is quantized to 3-bit and 2-bit representations. For the baseline LSQ+ quantization (Bhalgat et al., 2020), the attention substantially deteriorates and distributes uniformly across the given input when quantized to extremely low bit-widths. However, our 2-bit quantized Swin-T is still capable of segmenting the salient object region effectively.

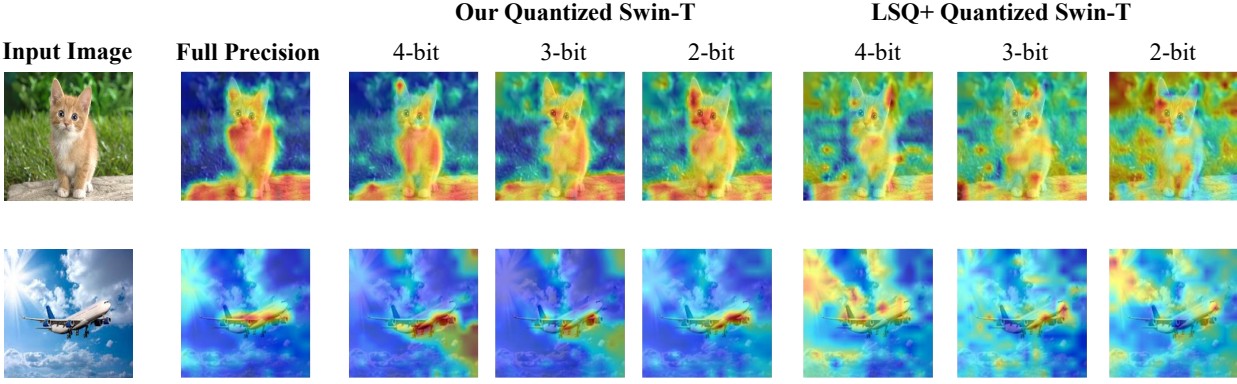

Figure 7: The comparison of attention map visualization of quantized Swin-T using our method and LSQ+ (Bhalgat et al., 2020).

## 5 Conclusion

In this work, we have provided a comprehensive understanding of the complexities associated with Vision Transformers quantization. Through an in-depth analysis of quantization sensitivity, and contrasting CNNs with ViTs, we elucidate that the **variation** behavior inherent to ViTs poses considerable challenges to quantization-aware training. Specifically, the variation in ViTs can induce oscillatory phenomena, necessitating an extended convergence period due to the consequent instability. To address the challenges presented by variation, we propose an effective variation-aware quantization technique. The multi-crop knowledge distillation strategy enhances accuracy and efficiency by mitigating the variation within the mini-batch. Furthermore, we introduce module-dependent quantization and oscillation-aware bin regularization to ensure that the optimization process remains unaffected by variation and to suppress the oscillatory effect instigated by variation. Through extensive demonstrations, we have shown that our proposed solution to variation in ViTs results in state-of-the-art accuracy on the ImageNet-1K dataset across various ViT architectures.

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

# A    Appendix

When comparing the results of ViT quantization using our methods with other methods in the experiments, we use the training settings and hyper-parameters shown in Table 7. Generally, most of these hyper-parameters and training settings are the same across different ViT models and different bitwidths settings. We found that applying the proposed Oscillation-aware Bin Regularization (OBR) is more effective for low-bit quantization, including 3-bit and 2-bit. The different performance of OBR among different bitwidth is mainly because penalizing the oscillation during QAT will harm the normal optimization of latent weights, which is more prominent in higher bitwidth. Accordingly, we only apply OBR to the 2-bit and 3-bit quantization.

Table 7: Detailed hyper-parameters and training scheme for different ViT architectures.

| Network | DeiT-T | SReT-T | Swin-T |
|---|---|---|---|
| Epoch | 150 | 150 | 150 |
| Batch Size | 1024 | 640 | 512 |
| Teacher | EfficientNet-L2 (Xie et al., 2020) | EfficientNet-L2 | EfficientNet-L2 |
| Optimizer | AdamW | AdamW | AdamW |
| Initial $lr$ | 5e-4 | 5e-4 | 5e-4 |
| $lr$ scheduler | Consine | Consine | Consine |
| Min $lr$ | 1e-5 | 1e-5 | 5e-6 |
| Warmup $lr$ | 1e-6 | 1e-6 | 1e-6 |
| Weight decay | 1e-4 | 1e-4 | 1e-4 |
| Warmup epochs | 5 | 5 | 5 |
| Random Resize & Crop | ✓ | ✓ | ✓ |
| Random Horizontal Flip | ✓ | ✓ | ✓ |
| Color jittering | - | - | - |
| Number of Crops | 4 | 4 | 4 |

Fig. 8 compares the training loss and Top-1 test accuracy for 4-bit quantized DeiT-T using our method and LSQ+ (Bhalgat et al., 2020). The core advantages of both effectiveness and efficiency are shown here. In terms of effectiveness, our method can achieve higher Top-1 accuracy and has a more stable loss scheme. For efficiency, our method helps the model converge faster, with only half of the total training epochs.

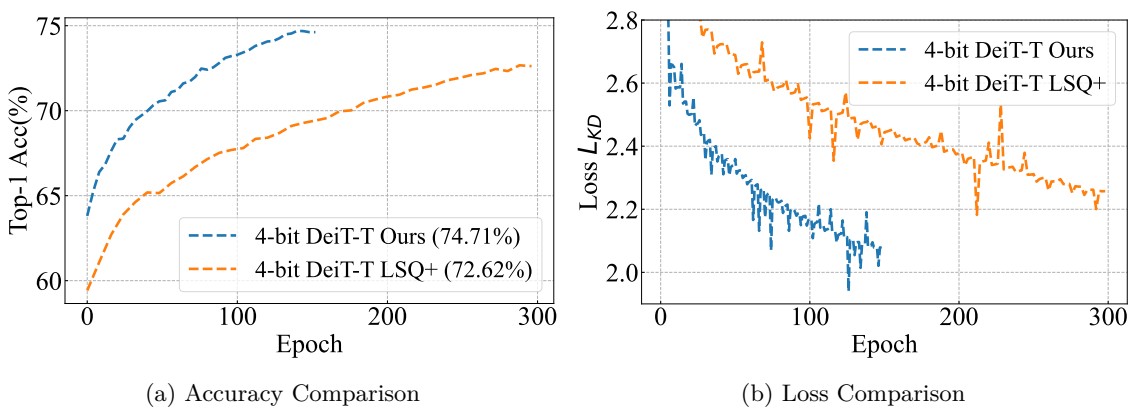

(a) Accuracy Comparison                    (b) Loss Comparison

Figure 8: Training dynamics of 4-bit quantized DeiT-T with our methods and LSQ+ (Bhalgat et al., 2020).

