# OpenReview forum: "Variation-aware Vision Transformer Quantization"
_TMLR — Rejected by TMLR_

### Review · Reviewer_CQfo · 2023-07-16

**Summary Of Contributions:**

This paper addresses the problem of weight quantization and pruning of vision transformers. These models have shown effective results in computer vision, but also have a significantly large parameter and FLOP count. Therefore, quantization and restraining approaches are necessary for computation-constrained applications. The paper exhibits multiple analyses of why current quantization approaches don't carry over to vision transformers. A module-dependent quantization is proposed, and experimental results show that this improves classification accuracy performance at low quantization settings.

**Audience:**

Yes

**Broader Impact Concerns:**

Not applicable. The paper addresses quantization approaches which could improve accessibility of these methods and models.

**Claims And Evidence:**

Yes

**Requested Changes:**

There are a few inconsistencies in the details that are confusing:

* Table 4 and Table 6 are incomparable as the experiments were run using different base models?

* Figure 8, what is the reason that the proposed method finishes so much faster? Likewise, in Table 3 it seems that the new method trains on half as many epochs, is that a feature or a bug?

* Equation 11 and Table 6: what does f represent, and can one assume that its result is reported in the final column in Table 6? I do not understand how this number reflects oscillation, as EMA({0,0,0,0}) will be similarly low as EMA({+1,-1,+1,-1}).

**Strengths And Weaknesses:**

Strengths:

* The paper proposes various examples and potential explanations for why previous quantization techniques for CNNs don't easily carry over to ViT models. For example, Table 1 studies the modules that suffer most from quantization; Figure 2 studies which layer suffers most from quantization of weights.

Weaknesses:

* For a stronger argument in Table 1, one could also study leave-one-out quantization. This table studies quantizing all modules except one, and reasons that MHSA yields the largest increase in accuracy. Additionally, one could quantize only one module and study which module comes with the largest performance decrease.

* A loss function is proposed to counteract the oscillatory behaviour of weights during training. This effect was empirically shown in Figure 4b. However, what is the reason to assume that removing the oscillation might improve the accuracy? Also, in the experimental result section, I don't see evidence that the OBR loss actually improves something. Moreover, Table 4 seems to suggest that only a half accuracy point is lost when not running this loss function, and these results are generally lower than Table 6.

* Figure 6: what is on the x-axis of this Figure, and how are the axes chosen? According to the introduction, these models have millions of weights, so it's hard to draw a conclusion from observing only two axes of variation. Moreover, the method for choosing these axes might change the conclusion.

---

> ### Author Response · Authors · 2023-08-12
> **Rebuttal by Authors (1/2)**
>
> We thank the reviewer for the insightful comments and valuable suggestions. Please see our responses and clarifications to the concerns below.
>
> Q1:  For a stronger argument in Table 1, one could also study leave-one-out quantization. This table studies quantizing all modules except one, and reasons that MHSA yields the largest increase in accuracy. Additionally, one could quantize only one module and study which module comes with the largest performance decrease.
>
> A1: Thanks for pointing this out. We carry out the “leave-one-out quantization” experiment as we believe that by quantizing every module except one, we can more precisely estimate the real sensitivity of each module to quantization. In practical quantization-aware training situations, the majority of the modules operate in low precision and are interconnected. Additionally, this form of analysis has previously been explored in the study Q-ViT [1]. The “quantize-one-module-only” experiments are shown as follows, which also proves that MHSA is the most sensitive module to the quantization perturbation. We will refer to the mentioned work and include the additional experiments in our revision.
>
> | Quantization Target | Top-1 Acc(%) | Top-5 Acc(%) | Quantized Para(%) |
> |---------------------|--------------|--------------|-------------------|
> | None (FP Model)     | 73.75        | 91.78        | 0                 |
> | All (Baseline 3-bit)| 68.22        | 88.56        | 100               |
> | FFN only            | 73.51        | 91.72        | 62.1              |
> | MHSA only           | **72.90**    | **91.29**    | 31.1              |
> | _query_ in MHSA only| 73.32        | 91.55        | 7.8               |
> | _key_ in MHSA only  | **73.18**    | **91.40**    | 7.8               |
> | _value_ in MHSA only| 73.38        | 91.53        | 7.8               |
>
> Q2:  A loss function is proposed to counteract the oscillatory behaviour of weights during training. This effect was empirically shown in Figure 4b. However, what is the reason to assume that removing the oscillation might improve the accuracy? Also, in the experimental result section, I don't see evidence that the OBR loss actually improves something. Moreover, Table 4 seems to suggest that only a half accuracy point is lost when not running this loss function, and these results are generally lower than Table 6.
>
> A2: Thanks for the comments. We would like to clarify that Table 4 presents the experiment for 4-bit DeiT-T, whereas Table 6 details the results of 3-bit SReT-T. Directly comparing these results might not be accurate. Our hypothesis that oscillation adversely affects accuracy stems from the fact that the latent weights tend to situate at the decision boundary between quantization bins. Such placement deviates from the optimal distribution and convergence. This observation aligns with findings from a previous study by Nagel et al. [2]. We will incorporate more analysis linking oscillation to the quantized model's performance in the revision.
>
> Q3:  Figure 6: what is on the x-axis of this Figure, and how are the axes chosen? According to the introduction, these models have millions of weights, so it's hard to draw a conclusion from observing only two axes of variation. Moreover, the method for choosing these axes might change the conclusion.
>
> A3: Thanks for the question. We follow Loss Landscape [3] for the visualization in Figure 6. The axes are not specific weights or parameters but are two random direction vectors $\delta$ and $\eta$. We will apply a perturbation with these two vectors equally on all parameters $\theta$ and plot the loss $f(\alpha, \beta)=L(\theta+\alpha\delta+\beta\eta)$ as the z-axis. The random direction vectors $\delta$ and $\eta$ are sampled from a random Gaussian distribution with appropriate scaling, which ensures that the plotted loss landscape is similar across different direction vectors.
>
> Q4: Table 4 and Table 6 are incomparable as the experiments were run using different base models?
>
> A4: Thanks for the suggestion. The experiments of Table 4 are conducted on DeiT-T, and experiments of Table 6 are conducted on SReT-T. Here we provide an updated Table 6 of the regularization experiments on 4-bit quantized DeiT-T, and this will also be included in our revised paper.
>
> | Regularization           | Top-1 Acc   | Top-1 Acc   | Oscillation |
> |--------------------------|-------------|-------------|-------------|
> | Baseline                 | 74.22       | 91.41       | 6.98        |
> | KURE                     | 73.79       | 91.15       | 7.30        |
> | Ours, $\lambda$=cos(0,1) | 74.64       | 91.69       | 0.19        |
> | Ours, $\lambda$=cos(0,0.1)| 74.71       | 92.02       | 0.54        |
> | Ours, $\lambda$=cos(0,0.01)| 74.35     | 91.50       | 3.20        |

---

> ### Author Response · Authors · 2023-08-12
> **Rebuttal by Authors (2/2)**
>
> Q5: Figure 8, what is the reason that the proposed method finishes so much faster? Likewise, in Table 3 it seems that the new method trains on half as many epochs, is that a feature or a bug?
>
> A5: Thanks for pointing this out, and this is an advantage of our method with faster convergence.
> This superiority over other methods can be attributed to our approach in addressing the oscillation issues inherent in quantization-aware training, which often leads to slower and less optimal convergence. We are confident that our proposed MCKD not only stabilizes the training but also significantly expedites the convergence of our quantized model.
>
> Q6:  Equation 11 and Table 6: what does f represent, and can one assume that its result is reported in the final column in Table 6? I do not understand how this number reflects oscillation, as EMA({0,0,0,0}) will be similarly low as EMA({+1,-1,+1,-1}).
>
> A6: Thanks for pointing this out. Here, $f$ represents the estimated frequency of the oscillation over different training iterations $t$. When $f^t > 0.005$, we define the corresponding target weights as oscillating. The percentage of oscillation in Table 6 conveys the proportion of weights where $f^t > 0.005$ at the end of the training. This provides insight into oscillation, especially as our primary focus is not the weight itself but the sign of the weight update (it's essential to note that the sign can never be zero). As claimed in Equation 10, we define weight to be oscillating only when the integer value changes and the update directions change. For Equation 11, we would like to correct it to the following to avoid confusion:
>
> $f^t=m\cdot o^t + (1-m)\cdot f^{t-1}$,
>
> where $o^t$ is a boolean indicator of  $o^t$ = $(x_t^{int}$ $\neq$ $x_{t-1}^{int})$ $\text{AND}$ $(\text{sign}(\Delta^t_{\text{int}}) \neq \text{sign}(\Delta^{t-1}_{\text{int}}))$
>
> If the weight does not oscillate, the sign of update is fixed, for example, ${0, 0, 0, 0}$. While for the weights in oscillation, the sign will change from time to time at most iterations, for example, ${+1,+1,0,+1}$. The EMA({+1,+1,0,+1}) will be significantly higher than EMA({0, 0, 0, 0}), which reflects the oscillation.
>
> [1] Li, Zhexin, et al. "Q-vit: Fully differentiable quantization for vision transformer." arXiv preprint arXiv:2201.07703 (2022).
>
> [2] Nagel, Markus, et al. "Overcoming oscillations in quantization-aware training." International Conference on Machine Learning (ICML) 2022.
>
> [3] Li, Hao, et al. "Visualizing the loss landscape of neural nets." Advances in Neural Information Processing Systems (NeurIPS) 2018

---

### Review · Reviewer_qHCU · 2023-07-22

**Summary Of Contributions:**

This paper proposes a variation-aware quantization method. The author first analyzes the variation behavior in ViTs. Then the author proposes multi-crop knowledge distillation strategy, a module-dependent quantization scheme, and a regularization strategy.

**Audience:**

No

**Claims And Evidence:**

No

**Requested Changes:**

1.	The paper should give a more explanation for the mentioned weakness.

**Strengths And Weaknesses:**

Strength:
1. The target issues of the paper are meaningful and worth exploring. The motivation is clear.
2. This submission gives a valuable implementation of such an idea.

Weakness:
1. In Figure 1, this paper does not compare with some SOTA methods. The compared method is published in 2020.
2. In Table 3, this paper only compares three methods. More SOTA methods should be compared and should analyze what leads to their performance difference.
3. The idea is not new and very simple. The idea of multi-crop can be seen in the distillation area.

---

> ### Author Response · Authors · 2023-08-12
> **Rebuttal by Authors**
>
> We thank the reviewer for the constructive feedback and detailed comments. Please see our responses and clarifications to the concerns below.
>
> Q1:  In Figure 1, this paper does not compare with some SOTA methods. The compared method in Figure 1 is published in 2020.
>
> A1: Thanks for the suggestion. Our method also outperforms SOTA methods Q-ViT (2022) and Li et al. (2022) as shown in Table 3. It's important to mention that since the Top-1 accuracy of the baseline full-precision models are different among these SOTA methods, for a fair comparison, we solely visualize the accuracy of the quantized vision transformer using LSQ+. We will include the accuracy curve of Q-ViT in Figure 1 in the revision and also mark the full-precision model accuracy of different methods.
>
> Q2:  This paper only compares three methods in Table 3, more SOTA methods should be compared and should analyze what leads to their performance difference.
>
> A2: Thanks for the suggestion. Our approach aligns with the quantization-aware training (QAT) methodology. In contrast, the majority of prior ViT quantization techniques predominantly adopt the post-training quantization (PTQ) strategy. It's important to note that these methods aren't directly comparable to ours. We've elaborated on this in the “quantization techniques” part of the related work section. More recently, a few QAT-centric ViT quantization methods have emerged as [1, 2]. We've provided an extended comparison of these methods as shown in the table below, which will be integrated into the revised experimental section.
>
> |  Network   | Method  |  FP Top-1 |  Bit-width  | Top-1 |
> |  ----  | ----  | ----  | ----  | ----  |
> | DeiT-T  | I-ViT [1]             | 72.21 | 8/8 | 72.24|
> | DeiT-T  | GPUSQ-ViT [2] | 72.2   | 4/4 | 71.7  |
> | DeiT-T  | Ours                 | 72.86 | 4/4 | 74.71|
> | Swin-T  | I-ViT[1]             | 81.35 | 8/8 | 81.5  |
> | Swin-T  | GPUSQ-ViT [2] | 81.2   | 4/4 | 80.7  |
> | Swin-T  | Ours                 | 81.0   | 4/4 | 82.42|
>
> The results showcasing how our method surpasses current SOTA methods have been elaborated in section 4.3 of “Ablation Study”. In essence, while earlier methods struggled to address and minimize the variation within ViT, our approach successfully harnesses Multi-crop Knowledge Distillation (MCKD), Module-dependent Quantization (MDQ), and Oscillation-aware Bin Regularization (OBR) to effectively suppress such variations.
>
>
> Q3:  The idea is not new and very simple. The idea of multi-crop can be seen in the distillation area.
>
> A3: Thanks for the comments. We clarify this study introduces a unique and novel approach by delving into the inherent challenges associated with ViT quantization and pinpointing its distinct variation tendencies. While we acknowledge in the introduction's contribution section that the multi-crop knowledge distillation isn't a groundbreaking method, our exploration into its impact on variance reduction is pioneering. We believe that this analysis furnishes valuable insights for effective ViT quantization.
>
> [1] Li, Zhikai, and Qingyi Gu. "I-ViT: integer-only quantization for efficient vision transformer inference." arXiv preprint arXiv:2207.01405 (2022).
>
> [2] Yu, Chong, et al. "Boost Vision Transformer with GPU-Friendly Sparsity and Quantization." Proceedings of the IEEE/CVF Conference on Computer Vision and Pattern Recognition. 2023.

---

### Review · Reviewer_tq5R · 2023-08-01

**Summary Of Contributions:**

This paper initially highlights the phenomenon of variation in Vision Transformers and introduces a variation-aware quantization method rooted in knowledge distillation.

**Audience:**

Yes

**Claims And Evidence:**

Yes

**Requested Changes:**

Na

**Strengths And Weaknesses:**

Strengths:

The paper is well written and articulates its points effectively.

The inclusion of experimental analysis on vision transformers is a strength of the paper. This analysis provides valuable insights and promotes understanding of the proposed method.

The visualizations provided in Figures 6 and 7 are beneficial in enhancing readers' comprehension of the methodology.

Weaknesses :

The paper's emphasis on the phenomenon of training oscillation in vision transformers is not novel. This issue has been previously addressed in multiple studies.

The scope of the experiments is limited. The author focused only on a small scale Vision Transformer (ViT) model. It would be beneficial to extend these experiments to other ViT models (e.g., Base, Large) or MAE-Base, Large models. This expansion would provide a broader perspective on the effectiveness of the proposed method.

The proposed Multi-crop Knowledge Distillation seems to have been previously introduced in another paper [1]. Moreover, the concept of knowledge distillation aiding the optimization of quantized model training has been studied in prior works [2]. This suggests that the novelty of the proposed method may be questionable.



References:
[1] https://arxiv.org/pdf/2112.01528.pdf


[2] Model compression via distillation and quantization

---

> ### Author Response · Authors · 2023-08-12
> **Rebuttal by Authors**
>
> We thank the reviewer for the detailed comments and the precise summarization of our work. Please see our responses and clarifications to the concerns below.
>
> Q1:  The paper's emphasis on the phenomenon of training oscillation in vision transformers is not novel. This issue has been previously addressed in multiple studies.
>
> A1: Thanks for pointing this out. We have delved into the oscillation effect and have also discussed previous works in Section 3.2.3. Our primary objective, distinct from prior research, is to demonstrate that Vision Transformers are more prone to oscillation than ConvNets. *This novel insight* has not been highlighted in earlier methods. Additionally, we furnish an analysis pinpointing the underlying causes of oscillation, framed in terms of variation. This can subsequently be addressed using our proposed framework.
>
> Q2:  The scope of the experiments is limited. The author focused only on a small scale Vision Transformer (ViT) model. It would be beneficial to extend these experiments to other ViT models (e.g., Base, Large) or MAE-Base, Large models. This expansion would provide a broader perspective on the effectiveness of the proposed method.
>
> A2: Thanks for the valuable suggestion. We have conducted additional experiments on a larger ViT model DeiT-S. The results are listed as follows and will be included in our revision.
>
>
> |  Network   | Method  | Epochs |  FP Top-1 |  Bit-width  | Top-1 | Bit-width  | Top-1 |
> |  ----  | ----  | ----   | ---- | ----  | ----  | ----  | ----  |
> | DeiT-S  | Q-ViT  | 300 | 79.05  | 4/4 | 79.75 | 3/3 | 77.76 |
> | DeiT-S  | LSQ+  | 300 | 79.05  | 4/4 | 78.91 | 3/3 | 77.24 |
> | DeiT-S  | Ours    | 150 | 79.05  | 4/4 | 80.12 | 3/3 | 79.02 |
>
>
> Q3:  The proposed Multi-crop Knowledge Distillation seems to have been previously introduced in another paper [1]. Moreover, the concept of knowledge distillation aiding the optimization of quantized model training has been studied in prior works [2]. This suggests that the novelty of the proposed method may be questionable.
>
> A3: Thanks for your question and the references. In the introduction's contribution section, we referenced [1]. While we are not introducing the Multi-crop Knowledge Distillation scheme, we are leveraging it to minimize variation within mini-batches, thereby enhancing our quantized ViT's performance. We concur that the concept of quantization with knowledge distillation isn't groundbreaking and isn't the key of our work. Our primary contribution lies in harnessing distillation to diminish variations and shed light on the mechanisms and rationale behind the enhanced performance of the quantized model.

---

### Decision · Action_Editors · 2023-10-10

**Recommendation:** Reject

**Comment:**

The paper is promising to be potentially accepted given the listed strengths and the unanimous recommendation from the reviewers. However, a major revision before the acceptance is needed:

1. Section 3.3.1 needs to be rewritten. The way it's currently presented is confusing - as if multi-crop KD is proposed by this paper. This is correctly pointed out by Reviewer [tq5R]. This detailed information and connection was missing in Section 3.3.1 even though the authors did cite the related paper. The authors should convey the message more clearly in Section 3.3.1 and the rest of the paper.

2. Please add the additional experiments for ViT/MAE-base and ViT/MAE-large as suggested by Reviewer [tq5R].

3. Please include the additional results from rebuttal into the revised paper and make the adjustments requested by Reviewer [CQfo].

4. The paper emphasizes the contribution of finding larger weight distribution variance in ViT quantization and the accordingly proposed module-dependent quantization. The overall position of the paper is that the findings and proposed methods are ViT-specific, and are thus valuable to the ViT community. However, adding more degree of freedom to scaling factors would improve the quantization performance is not surprising. Is this truly a ViT-specific thing (caused by the observation), or it's more because of the additionally introduced bits? Additional comparison to CNNs is needed to verify the claim. Especially, for certain CNN architectures such as ResNeXt, one has a set of different transformations similar to MHSA. Comparison and additional observations (e.g., weight distribution variance) on these architectures would make the claim more convincing.

The above changes are considered to be non-trivial and would require a major revision of the paper.

**Audience:**

Yes, the efficiency-aware ML and general visual recognition communities will be interested in this paper.

**Claims And Evidence:**

This paper studies the unique variation behaviors of ViT quantization and accordingly proposes an efficient knowledge-distillation-based
variation-aware quantization method. Multi-crop knowledge distillation is used to accelerate and stabilize the training and alleviate the variation’s influence during QAT. The authors also propose a module-dependent quantization scheme and a variation-aware regularization term to suppress the oscillation of weights.

This is a borderline paper. The paper is in general well written and all reviewers acknowledge the strengths of the paper. Reviewer [tq5R] and [CQfo] consider the ViT-specific findings in quantization interesting. Reviewer [qHCU] initially had concerns on the lack of comparison to SOTA methods. The concerns were addressed by rebuttal and Reviewer [qHCU] considers the result promising.

The paper still lacks enough validation in certain aspects, as detailed in the Comment Section.

**Resubmission Of Major Revision:**

The authors may consider submitting a major revision at a later time.